# Modelling the Behaviour of Currency Exchange Rates with Singular Spectrum Analysis and Artificial Neural Networks

**Paulo Canas Rodrigues** [1],[*] , **Olushina Olawale Awe** [1],[2] **, Jonatha Sousa Pimentel** [1] **and Rahim Mahmoudvand** [3]

[1]   Department of Statistics, Federal University of Bahia, Salvador 40170-110, Brazil; oawe@aul.edu.ng (O.O.A.); jsppimentel9@gmail.com (J.S.P.)
[2]   Department of Mathematical Sciences, Anchor University Lagos, Lagos 100278, Nigeria
[3]   Department of Statistics, Bu-Ali Sina University, Hamedan 6517833131, Iran; r.mahmodvand@gmail.com
[*]   Correspondence: paulocanas@gmail.com

**Abstract:** A proper understanding and analysis of suitable models involved in forecasting currency exchange rates dynamics is essential to provide reliable information about the economy. This paper deals with model fit and model forecasting of eight time series of historical data about currency exchange rate considering the United States dollar as reference. The time series techniques: classical autoregressive integrated moving average model, the non-parametric univariate and multivariate singular spectrum analysis (SSA), artificial neural network (ANN) algorithms, and a recent prominent hybrid method that combines SSA and ANN, are considered and their performance compared in terms of model fit and model forecasting. Moreover, specific methodological and computational adaptations were conducted to allow for these analyses and comparisons.

**Keywords:** singular spectrum analysis; multivariate singular spectrum analysis; time series forecasting; artificial neural networks; currency exchange rates

---

## 1. Introduction

Apart from other important economic indicators such as interest rates, consumer price index, money supply and inflation, the currency exchange rate is one of the most important determinants of a country's relative level of economic health [1]. Exchange rates play a vital role in any country's level of trade, which is critical to every free market economy in the world [2,3]. No economy can operate in autarky, therefore, exchange rates are among the most analysed and governmentally manipulated economic indicators in any nation. Recently, exchange rates forecasting have become an important economic problem that is receiving increasing attention among researchers and policy makers especially because of its practical national economic significance. A fluctuating (volatile) exchange rate might lead to an unstable economy where it becomes difficult to predict the value of goods, services and other important economic components. Exchange rates have been shown, in the literature, to be among the major challenging and difficult economic measures to accurately forecast because changes in exchange rates are erratic and can have drastic effects on the economy [4–6]. Erratic behaviour of exchange rate was also identified in the literature as part of the leading causes of economic recessions [7]. Various nations adopt different exchange rate systems based on their history and economic goals. For instance, Brazil, India and South Africa implement a free floating exchange rate system while China and Russia adopts a system of managing floating exchange rates.

According to the International Monetary Fund (IMF), the countries that form the BRICS nations (Brazil, Russia, India, China, and South Africa) have more than 25% of the world's land and 40%

of the world's population and about 18.3% of global nominal output [8]. The exchange rate, as the main system for foreign exchange of a country, has become a key factor affecting the stable economic development of the BRICS countries. The BRICS nations are the fastest growing in the emerging economies of the world. However, in recent years, the exchange rates of these economies have all experienced periods of high volatility [8]. In the context of the gradual recovery of the US economy and the relatively poor economic situations in Europe, United Kingdom and Japan, these emerging economies have been experiencing relatively stable economic growth except for recent unpalatable global circumstances. This article is therefore based on the examination of a suitable forecasting model for predicting currency exchange rates with special emphasis on the BRICS nations. Here, besides the currency exchange rates of the BRICS currencies with respect to the United States dollar (USD), we also consider three other powerful world currencies: the British pound (GBP), the Euro (EUR), and the Japanese yen (JPY).

The main objective of this paper is to access the performance of classical and contemporary methods for model fit and model forecasting in currency exchange rates. In particular, we want to compare the success of recently proposed hybrid methods with classical parametric and non-parametric, univariate and multivariate methods in the context of currency exchange rates. To achieve our objectives, we consider daily exchange rates data consisting 4240 observations each for eight currencies from 01/12/2003 to 28/02/2020 and employ time series techniques such as the autoregressive integrated moving average (ARIMA) model, the non-parametric univariate and multivariate singular spectrum analysis (SSA), artificial neural network (ANN) algorithms, and an hybrid method that combines the SSA with the ANN. Moreover, we adapt the hybrid method for model fit as it was originally proposed for model forecasting only [9]. Comparisons are made for model fit and model forecasting by employing the root mean square error (RMSE) and the mean absolute percentage error (MAPE).

The rest of this paper is structured as follows: Section 2 presents the contextual issues, the models and methodologies about the considered models for time series model fit and model forecasting. Section 3 presents the empirical results and discussions, and Section 4 gives a short discussion and concludes the paper.

## 2. Materials and Methods

In this section, we present the data used in this study and give a brief description of the forecasting models employed in this article.

### 2.1. The Data

This study employs data on daily exchange rate of eight currencies, in reference to the United States dollar (USD), spanning seventeen years between 1 December 2003 and 28 January 2020 (4240 observations). These data were obtained from www.yahoo.finance. The currencies analysed and compared (Figure 2) are: Brazilian real (USD/BRL), Russian rouble (USD/RUB), Indian rupee (USD/INR), Chinese renminby (USD/CNY), South African rand (USD/ZAR), British pound (USD/GBP), Euro (USD/EUR), and Japanese yen (USD/JPY).

### 2.2. Autoregressive Integrated Moving Average (ARIMA) Model

In time series analysis, an autoregressive integrated moving average (ARIMA) model is a generalization of an autoregressive moving average (ARMA) model. Both of these models are fitted to time series data either to better understand the data or to predict future points in the series (forecasting). The auto regressive (AR) part of ARIMA indicates that the evolving variable of interest is regressed on its own lagged or prior values. The moving average (MA) part indicates that the regression error is actually a linear combination of error terms whose values occurred contemporaneously and at various times in the past. The "integrated" (I) part of the ARIMA model indicates that the data values were replaced with the difference between the data values and their previous values [10]. This parametric

model can then be written as $ARIMA(p, d, q)$, with $p$, $d$ and $q$ non-negative integers [11]. Given a time series $Y = y_1, \ldots, y_N$, the $ARIMA(p, d, q)$ model can be written as:

$$(1 - \phi_1 B^1 - \cdots - \phi_p B^p)(1 - B)^d y_t = c + (1 + \theta_1 B^1 + \cdots + \theta_q B^q)\varepsilon_t, \tag{1}$$

where $\phi_1, \ldots, \phi_p$ are the parameters or coefficients of the $p$ autoregressive terms; $B$ is the time lag operator, or backward shift, which is a linear operator denoted by $B^k$ such that $B^k y_t = y_{t-k}$, $t \in \mathbb{Z}$; $y_t$ the observation at the time point $t$; $c = \mu(1 - \phi_1 - \cdots - \phi_p)$; $\mu$ is the mean of $(1 - B)^d y_t$; $\beta_1, \ldots, \beta_q$ are the parameters or coefficients of the $q$ moving average terms; and $\varepsilon_t$ is an error term, usually white noise with variance $\sigma^2$. The results presented in this paper are based on an alternative parametrization of the ARIMA model that is implemented in the `arima` function of the software R [12].

In this study, we only consider the classical ARIMA-based models from the class of pure parametric models. However, nonparametric and ANN-based approaches are also considered. In a recent study by [13] the supremacy of ANN over ARIMA or generalized autoregressive conditional heteroskedasticity (GARCH) model for time series prediction was discussed. On the other hand, Ref. [14] compared the methods of ARIMA, ANN and fuzzy systems on 1284 daily observations of seven major currencies for five years and concluded that ARIMA gives more significant results than ANN and fuzzy systems.

In the next subsection we briefly describe the ANN that is also considered in this paper.

### 2.3. Artificial Neural Network (ANN)

Neurons are the main cells that make up the nervous system and are responsible for conducting, receiving, and transmitting nerve impulses throughout the body, causing it to respond to stimuli in the environment, for example. The brain is a complex network of neurons that process information through a system of several interconnected neurons. It has always been challenging to understand brain functions; however, due to advances in computing technologies, we can now program neural networks artificially [15].

Neural networks were originally developed in cognitive science and later used in engineering for pattern recognition and classification [16]. Neural networks are particularly useful because they can be used to model nonlinear behaviour in economics and financial markets, in contrast to traditional linear models which are more restrictive. They also have the capability of being able to approximate any nonlinear function and decompose "noisy" data. They proved, in some instances, to be more effective in describing the dynamics of nonstationary time series due to their unique nonparametric, noise-tolerant, and adaptive properties [17]. Over the last few decades, researchers and practitioners alike showed growing interests in applying modified versions of ANNs for time series analysis and forecasting [18]. ANNs are an effective tool to realize any nonlinear input-output mapping. It was demonstrated that with a sufficient number of hidden layer units, an ANN is capable of approximating any continuous function to any desired degree of accuracy [17]. Due to the nature of their learning process, ANNs can be regarded as nonlinear autoregressive models [19].

Artificial neural networks (ANNs) have gained tremendous popularity and use as a promising alternative technique for forecasting time series because of their several distinguishing features. The first networks developed were the Perceptron and Adaline networks, developed in the 1950s and 1960s by Rosenblatt [20] and Widrow [21] respectively. Perceptron networks were developed with the objective of recognizing images, being a model that received a set of input data and returned a single binary output. Adaline networks were developed to be used for pattern recognition, signal processing and regression. Similar to the perceptron network in that it has several input layers and only one output, it differs in that its output is not binary but an activation function $f$.

Similar to the biological structure of neurons, artificial neural networks define the neuron as a central processing unit, which performs a mathematical operation that generates an output from a set of inputs [15]. The output of a neuron is a function of the weighted sum of the inputs plus the bias. The scheme of a simple artificial neural network can be seen in Figure 1.

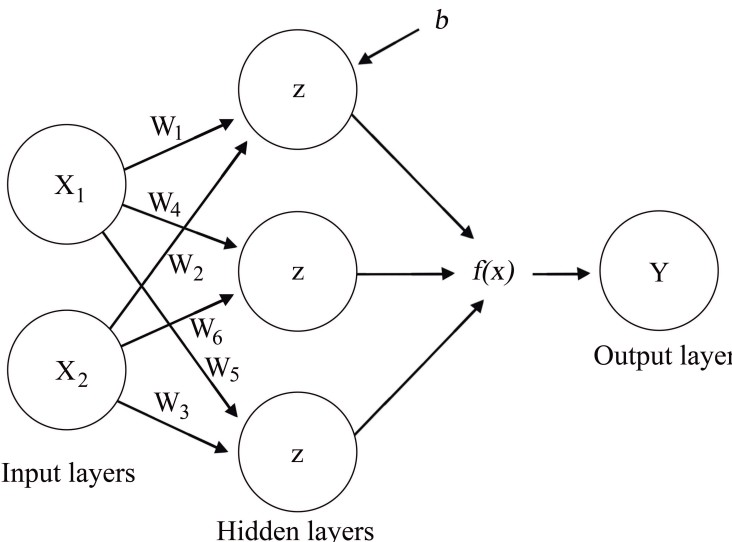

**Figure 1.** Representation of an artificial neural network.

An ANN is composed of the layers of input, output, and the so-called hidden layers, which are in the center of the network and with the help of so-called weights ($W_i$), bias ($b$) and the activation function $f$, converts the input data to the expected output. The weights in a neural network are the most important factor in the transformation of the input data at the output, functioning similarly to the functioning of the slope in linear regression. The weights here are numerical parameters that determine how strongly each neuron affects the other. Meanwhile, the bias is like the intercept added in a linear equation, being an additional parameter that is used to adjust the output together with the weighted sum of the neuron inputs, and in each neuron there is still an activation process, through the $z$ function as

$$z = \sum_{l=1}^{L} W_l * X_l + b. \tag{2}$$

Finally, a function is applied to $z$, which is called the activation function $f$. The types of neurons are differentiated by the activation function attributed to them, and in practice the three most used functions are the sigmoid function, hyperbolic tangent and ReLU (rectified linear unit). There is also the loss function, which is the function used as a minimization criterion when estimating the parameters of a neural network. The most common loss function is the sum of squares of errors.

The neural network model for time series was applied with the aid of the R package `forecast`, through the `nnetar` function, which generates a feed-forward neural network with a single hidden layer and lagged inputs to forecasting univariate time series.

### 2.4. Singular Spectrum Analysis (SSA)

There is a vast literature on the non-parametric technique for time series modelling and forecasting SSA. SSA incorporates elements of classical time series analysis, matrix algebra, and multivariate statistics, and aims at decomposing a time series into a set of components that can be interpreted as trend components, seasonal and cyclic components and noise components [22–25]. This relatively new technique for time series analysis proved to be widely useful and applicable to many fields of application [9,26–38], with applications ranging from parameter estimation to time series filtering, and forecasting.

The basic SSA method consists of three complementary stages: decomposition, reconstruction and forecasting. The first stage is divided in two steps where the time series is decomposed in several components, in the second stage (two steps) the noise free time series is reconstructed and in the third

stage the reconstructed time series is used for out-of-sample forecasting. A short description of the SSA technique is given below. More information can be found in, e.g., [23–25,39].

### 2.4.1. First Stage: Decomposition

**1st step: Embedding.** Let $y_1, \ldots, y_N$ be a time series of length $N$. Considering a window length $L$ the result of this step is a $L \times K$ matrix $\mathbf{Y} = [Y_1 : \ldots : Y_K]$, where $K = N - L + 1$ and $Y_i = (y_i, \ldots, y_{i+L-1})^T$, $1 \leq i \leq K$.

**2nd step: Singular value decomposition (SVD).** In this step, the matrix $\mathbf{Y}$ will be decomposed using SVD as $\mathbf{Y} = \mathbf{Y}_1 + \cdots + \mathbf{Y}_L$, where $\mathbf{Y}_i = \sqrt{\lambda_i} U_i V_i^T$, $\mathbf{Y}_i = \mathbf{0}$ when $\lambda_i = 0$, and $V_i = \mathbf{Y}^T U_i / \sqrt{\lambda_i}$ with $\lambda_1, \ldots, \lambda_L$, the eigenvalues of $\mathbf{Y}\mathbf{Y}^T$ and $U_1, \ldots, U_L$, the corresponding eigenvectors.

### 2.4.2. Second Stage: Reconstruction

**3rd step: Grouping.** The grouping step corresponds to splitting the elementary matrices into $m$ disjunct subsets $I_1, \ldots, I_m$, and summing the matrices within each group. In our application we will focus on $m = 2$, i.e., only two groups. $I_1 = \{1, \ldots, r\}$ and $I_2 = \{r + 1, \ldots, L\}$ are associated with the signal and noise components, respectively.

**4th step: Diagonal averaging.** This step transforms each matrix $\mathbf{Y}_{I_j}$ into a new series of length $N$. Using diagonal averaging we have that $\mathbf{Y} = \widetilde{\mathbf{Y}}_{I_1} + \cdots + \widetilde{\mathbf{Y}}_{I_m}$, where $\widetilde{\mathbf{Y}}_{I_j}$ is the Hankelized form of $\mathbf{Y}_{I_j}$, $j = 1, \ldots, m$. Considering $\widetilde{y}_{m,n}^{(I_j)}$ the $(m, n)^{th}$ entry of the estimated matrix $\widetilde{\mathbf{Y}}_{I_j}$ and denoting by $\{\widetilde{y}_{j_1}, \ldots, \widetilde{y}_{j_N}\}$ the reconstructed components in the matrix $\widetilde{\mathbf{Y}}_{I_j}$, $j = 1, \ldots, m$, applying diagonal averaging follows that

$$\widetilde{y}_{j_l} = \begin{cases} \frac{1}{j_\ell - 1} \sum_{n=1}^{j_\ell - 1} \widetilde{y}_{n, j_\ell - n}^{(I_j)} & 2 \leq j_\ell \leq L - 1, \\ \frac{1}{L} \sum_{n=1}^{L} \widetilde{y}_{n, j_\ell - n}^{(I_j)} & L \leq j_\ell \leq K + 1, \\ \frac{1}{K + L - j_\ell + 1} \sum_{n=n-K}^{L} \widetilde{y}_{n, j_\ell - n}^{(I_j)} & K + 2 \leq j_\ell \leq K + L. \end{cases}$$

### 2.4.3. Third Stage: Forecasting

Two main algorithms for out-of-the-sample forecasting in the context of SSA are available: the recurrent SSA forecasting algorithm [23,40,41], and the vector SSA forecasting algorithm [23,42,43]. Here we will be interested in the recurrent SSA forecasting algorithm, which is briefly described below.

The basic requirement to obtain SSA out-of-sample forecasts is that the time series $Y_t = (y_1, \ldots, y_N)$ satisfies a linear recurrent formula, i.e., if a given observation can be written as a linear combination of the last $d$ observations:

$$y_t = a_1 y_{t-1} + a_2 y_{t-2} + \ldots + a_d y_{t-d}, \quad t = d + 1, \ldots, N. \tag{3}$$

Let us assume that $U_j^\nabla$ is the vector of the first $L - 1$ components of the eigenvector $U_j$ and $\pi_j$ is the last component of $U_j$ ($j = 1, \ldots, r$). Denoting $v^2 = \sum_{j=1}^{r} \pi_j^2$ we define the coefficient vector $\mathfrak{R}$ as:

$$\mathfrak{R} = \frac{1}{1 - v^2} \sum_{j=1}^{r} \pi_j U_j^\nabla.$$

Considering the above notation, the recurrent SSA forecasts $(\widehat{y}_{N+1}, \ldots, \widehat{y}_{N+h})$ can be obtained by

$$\widehat{y}_i = \begin{cases} \widetilde{y}_i, & i = 1, \ldots, N \\ \mathfrak{R}^T Z_i, & i = N + 1, \ldots, N + h \end{cases} \tag{4}$$

where $Z_i = [\widehat{y}_{i-L+1}, \ldots, \widehat{y}_{i-1}]^T$ and $\widetilde{y}_1, \ldots, \widetilde{y}_N$, are the SSA reconstructed values obtained from 4th step of the SSA algorithm described above.

### 2.4.4. SSA Parameter Selection

The SSA calibration depends on two parameters: the window length $L$, and the number of eigentriples used for reconstruction $r$. The choice of improper values for the parameters $L$ or $r$ yield incomplete reconstruction and the forecasting results might be misleading [41,43]. Despite the importance in choosing proper values for these parameters, no theoretical solution was proposed to solve this problem. An overall agreeable suggestion to choose the window length is to have it close to the middle of the series and proportional to the number of observations per period (e.g., to 12 for monthly time series, to four for quarterly time series, etc.). However, this choice does not guarantee the best predictions [41,43], being advisable a parameter choice made accordingly to the available data and intended analysis.

Among the alternative ways described in the literature to determine the number of eigentriples used for reconstruction $r$, the most widely used is the w-correlations approach. Considering two vectors $Y^{(1)} = [y_1^{(1)}, \ldots, y_N^{(1)}]^T$ and $Y^{(2)} = [y_1^{(2)}, \ldots, y_N^{(2)}]^T$, the w-correlation between them can be written as

$$\rho_w = \frac{\sum_{j=1}^{N} w_j^{L,N} y_j^{(1)} y_j^{(2)}}{\sqrt{\sum_{j=1}^{N} w_j^{L,N} \left(y_j^{(1)}\right)^2 \times \sum_{j=1}^{N} w_j^{L,N} \left(y_j^{(2)}\right)^2}}, \tag{5}$$

where $w_j^{L,N} = \min\{j, L, N - j + 1\}$ and $2 \leq L \leq N - 1$. According to this measure, two series (e.g., signal and noise components) are separable if the absolute value of their w-correlation is small. Therefore, we determine $r$ in such a way that the reconstructed series and residual have a small w-correlation between them. Another way to determine $r$ is by examining the forecast accuracy, i.e., $r$ is determined in such a way that the minimum error in forecasting will be obtained.

### 2.5. Multivariate Singular Spectrum Analysis (MSSA)

Multivariate SSA is a natural extension of the univariate SSA for analysing multivariate time series data. The algorithm is similar to the univariate SSA and has the same range of applications. Complete details about MSSA can be found in [23,39,44], and a brief description is presented below.

Let $Y_t = \left[y_t^{(1)}, \ldots, y_t^{(M)}\right]$, $t = 1, \ldots, N$, denote a sample of a $M$-variate time series with length $N$. Let us assume that $Y_t$ can be written in terms of a signal plus noise model as:

$$\mathbf{Y} = \begin{bmatrix} Y_1 \\ Y_2 \\ \vdots \\ Y_N \end{bmatrix} = \mathbf{S} + \mathbf{N} = \begin{bmatrix} s_1^{(1)} & \cdots & s_1^{(M)} \\ s_2^{(1)} & \cdots & s_2^{(M)} \\ \vdots & \cdots & \vdots \\ s_N^{(1)} & \cdots & s_N^{(M)} \end{bmatrix} + \begin{bmatrix} n_1^{(1)} & \cdots & n_1^{(M)} \\ n_2^{(1)} & \cdots & n_2^{(M)} \\ \vdots & \cdots & \vdots \\ n_N^{(1)} & \cdots & n_N^{(M)} \end{bmatrix}. \tag{6}$$

As with the univariate SSA, the goal here is to remove the noise, $\mathbf{N}$, from the original data and to obtain an estimate for the signal, $\mathbf{S}$, without having to specify a parametric form for the signal, which then can then be used to obtain out-of-the-sample forecasts. The MSSA algorithm also consists of three complementary stages just like the univariate case: decomposition, reconstruction and forecasting. In the first stage the series is decomposed; in the second stage the noise free series is reconstructed; and in the final stage the reconstructed time series is used to forecast new data points. Each stage in this algorithm includes two steps.

### 2.5.1. First Stage: Decomposition

**1st step: Embedding**. Considering the window length $L$, a full augmented trajectory matrix is constructed by a $L$-dimensional embedding of the time series with lag $l$, resulting in a block Hankel trajectory matrix $\mathbf{Y}$. Suppose $\mathbf{Y}^{(m)}, m = 1, \ldots, M$, denotes the Hankel matrix of dimension $L \times k$, $k = T - L + 1$, associated with the time series $m, m = 1, \ldots, M$. The trajectory matrix in MSSA can be defined as two different alternatives:

- **Horizontal form**:

$$\mathbf{Y} = \left[ \mathbf{Y}^{(1)}, \ldots, \mathbf{Y}^{(M)} \right] \tag{7}$$

- **Vertical form**:

$$\mathbf{Y} = \begin{bmatrix} \mathbf{Y}^{(1)} \\ \vdots \\ \mathbf{Y}^{(M)} \end{bmatrix}. \tag{8}$$

**2nd step: Singular value decomposition**. Let $\mathbf{U} = [U_1, \ldots, U_d]$ and $\mathbf{\Sigma} = \text{diag}\{\lambda_1, \ldots, \lambda_d\}$ denote the matrices with the eigenvectors and eigenvalues of $\mathbf{YY}'$, respectively. Then, we have $\mathbf{YY}' = \mathbf{U\Sigma U}'$ and $\mathbf{Y}$ can be decomposed by singular value decomposition as:

$$\mathbf{Y} = \mathbf{UU}'\mathbf{Y} = \sum_{i=1}^{d} U_i U_i' \mathbf{Y} = \mathbf{Y}_1 + \ldots + \mathbf{Y}_d, \tag{9}$$

where $\mathbf{Y}_i = U_i U_i' \mathbf{Y}$ is a unitary matrix corresponding to the $i^{\text{th}}$ largest singular value ($\sqrt{\lambda_i}$), and $d$ is the rank of $\mathbf{Y}$.

### 2.5.2. Second Stage: Reconstruction

**3rd step: Grouping**. Considering $\mathbf{Y}_i$ to be associated with the $i^{\text{th}}$ largest singular value of $\mathbf{Y}$, this step intends to separate the signal and noise components as follows:

$$\mathbf{Y} = \hat{\mathbf{S}} + \hat{\mathbf{N}}, \tag{10}$$

where $\hat{\mathbf{S}} = \mathbf{Y}_1 + \ldots + \mathbf{Y}_r$ and $r < d$ is the number of components associated with the signal.

**4th step: Diagonal averaging**. In this step, using anti-diagonal averaging on each block of $\hat{\mathbf{S}}$, the de-noised/smoothed time series will be reconstructed.

### 2.5.3. Third Stage: Forecasting

**5th step: Forecast engine**. The forecast engine of MSSA, which is a linear function of the last $L$ observations of the de-noised/smoothed time series, will be constructed in this step [39,44]. These forecasts are obtained by using the linear recurrent formula in a similar manner and detailed above for the univariate SSA algorithm. By considering the two versions of the trajectory matrix defined in the 1st step of this algorithm, we obtain the forecasts based on the horizontal MSSA (H-MSSA) and the forecasts based on the vertical MSSA (V-MSSA).

**6th step: Out-of-the sample forecasting**. In this step, $h$-steps ahead forecasts will be produced by using the forecast engine [39,44].

### 2.6. Hybrid Approach

To improve the results for model fit and model forecasting in time series many hybrid models, which combine more than one time series methodology, have been developed [9]. In some of those

cases, the SSA is first applied to the raw data in order to extract the deterministic component and then another method such as ANN is applied to the residuals of the SSA to fit/forecast the stochastic part of the time series [9,45]. In this analysis we will consider one of the methods proposed by [9] where the SSA, together with the recurrent SSA forecasting algorithm, is used to forecast the deterministic part of the series and an ANN is used to forecast the stochastic part associated with the signal left from the SSA fit. As with the methods presented before, this hybrid approach will be considered for both model fit and model forecasting.

### 2.7. Accuracy Measure

Here we will evaluate two types of errors: (i) in sample errors associated with model fit; and (ii) out-of-sample errors, associated with model forecasting. For each of the two types of errors, two measures will be considered: the RMSE, and the MAPE.

For model fit, the RMSE and MAPE are used as a criterion for accessing the quality of a model to fit the data, and it can be written, respectively, as:

$$RMSE = \sqrt{\frac{1}{N} \sum_{t=1}^{N} (y_t - \widetilde{y}_t)^2}, \tag{11}$$

and

$$MAPE = \frac{1}{N} \sum_{t=1}^{N} \left| \frac{y_t - \widetilde{y}_t}{y_t} \right|, \tag{12}$$

where $y_t$ are the observed values and $\widetilde{y}_t$ the fitted values by the considered model/algorithm (i.e., ARIMA, SSA, MSSA, ANN), and $N$ the length of the time series.

For model forecasting, let us assume that the last $g$ observations, e.g., $g = 12$, are used as the test set. The RMSE and MAPE to measure the out-of-sample forecasting error for a given model can be written, respectively, as:

$$RMSE = \sqrt{\frac{1}{g} \sum_{t=N-g+1}^{N} (y_t - \widetilde{y}_t)^2}, \tag{13}$$

and

$$MAPE = \frac{1}{g} \sum_{t=N-g+1}^{N} \left| \frac{y_t - \widetilde{y}_t}{y_t} \right|, \tag{14}$$

where $y_t$ are the last $g$ observed values and $\widetilde{y}_t$ the respective h-steps-ahead forecast values. Other measures such as the symmetric mean absolute percentage error or the mean directional accuracy can also be used to evaluate both model fit and model forecasting.

In this paper, we considered purely symmetric loss functions where the under-prediction and over-prediction of the currency exchange rates are considered to have the same importance. However, depending on the scope of the analysis, asymmetric loss functions that, e.g., give higher weights to losses of the currency exchange rates in relation to the USD, should be considered.

### 3. Results and Discussion

In this section, we will analyse the historical data from the eight currency exchange rates. This data will be used to compare: (i) the classical ARIMA model, (ii) the classical SSA algorithm, (iii) the classical MSSA algorithm, (iv) the artificial neural network algorithm, and (v) the hybrid algorithm that combines SSA and ANN, in terms of computational time and accuracy for model fit and model forecast. In terms of model assumptions, stationarity is of key importance. While many of the standard parametric time series methods (e.g., ARIMA) require the data to be stationary, the non-parametric SSA and MSSA do not require the this assumption in the data [23]. As for ANN, overfitting may ease the problem of having non-stationary time series significantly and might be a key to success for complex

financial time-series analysis [46]. The computational times presented in this section were obtained by a laptop with processor 2.00 GHz Intel Core i3-6006U, 4 GB RAM of memory and operational system of 64 bits with Windows 10.

Table 1 shows the descriptive statistics for the eight currency exchange rates, including the minimum, maximum, mean, standard deviation and coefficient of variation. Figure 2 shows the behaviour of the eight currency exchange rates along the time. From the analysis of these plots, it is possible to observe a different behaviour between series, and no clear pattern among developed or developing countries.

**Table 1.** Descriptive measures for the eight currency exchange rates.

| Currency | Minimum | Mean | Maximum | Standard Deviation | Coefficient of Variation |
|---|---|---|---|---|---|
| Brazilian real (USD/BRL) | 1.53 | 2.57 | 4.48 | 0.769 | 0.2992 |
| Chinese renminby (USD/CNY) | 6.03 | 6.93 | 8.28 | 0.691 | 0.0997 |
| Euro (USD/EUR) | 0.63 | 0.80 | 0.96 | 0.076 | 0.0951 |
| British pound (USD/GBP) | 0.47 | 0.64 | 0.83 | 0.090 | 0.1411 |
| Indian rupee (USD/INR) | 39.04 | 54.35 | 74.60 | 10.411 | 0.1916 |
| Japanese yen (USD/JPY) | 75.74 | 103.93 | 125.63 | 12.780 | 0.1230 |
| Russian rouble (USD/RUB) | 23.17 | 40.27 | 82.90 | 15.984 | 0.3969 |
| South African rand (USD/ZAR) | 5.60 | 9.71 | 16.87 | 3.050 | 0.3141 |

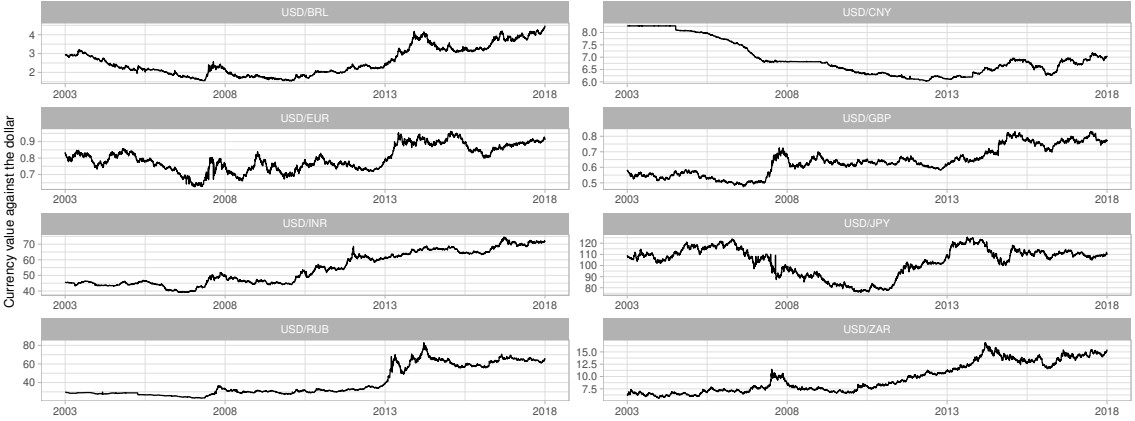

**Figure 2.** Time series for the exchange rates of the eight currencies against the USD. From top to bottom and from left to right: USD/BRL, USD/CNY, USD/EUR, USD/GBP, USD/INR, USD/JPY, USD/RUB and USD/ZAR. The vertical axes show the exchange rate and the horizontal axes shows the time.

### 3.1. Model Fit

The models/algorithms under comparison for model fit are: (i) ARIMA; (ii) SSA (with three alternative parameter choice); (iii) multivariate SSA (two algorithms); (iv) artificial neural networks (ANN); and (v) hybrid algorithm that combines SSA and ANN.

The parameters of the ARIMA model were estimated with the function "auto.arima" from the R package "forecast" [47], that does model selection based on either the Akaike information criterion or the Bayesian information criterion. The model parameters for the ARIMA models, together with the observed values of the test statistic and p-values of the Dicky-Fuller test (obtained using the function `adf.test` of the R package `tseries`) are given in Table 2. These results provide evidence that the stationarity requirement of the ARIMA model is met.

**Table 2.** Parameters for the ARIMA model, and observed valued of the test statistic and p-values for the Dickey-Fuller test.

| Currency | AR(p) | I(d) | MA(q) | Dickey-Fuller Test Test Statistic | p-Value |
|---|---|---|---|---|---|
| Brazilian real (USD/BRL) | 5 | 2 | 0 | $-13.586$ | 0.01 |
| Chinese renminby (USD/CNY) | 5 | 2 | 0 | $-13.189$ | 0.01 |
| Euro (USD/EUR) | 1 | 1 | 1 | $-15.531$ | 0.01 |
| British pound (USD/GBP) | 0 | 1 | 0 | $-15.420$ | 0.01 |
| Indian rupee (USD/INR) | 1 | 1 | 0 | $-15.313$ | 0.01 |
| Japanese yen (USD/JPY) | 0 | 1 | 1 | $-16.261$ | 0.01 |
| Russian rouble (USD/RUB) | 2 | 1 | 2 | $-14.292$ | 0.01 |
| South African rand (USD/ZAR) | 0 | 1 | 0 | $-16.945$ | 0.01 |

As mentioned above, for the SSA and multivariate SSA algorithms, there are two choices to be made by the researcher: (i) the window length $L$; and (ii) the number of eigentriples used for reconstruction $r$. The values for $L$ were chosen for each time series as defined in Table 3: $L_1 = N/20$, $L_2 = N/2$ and $L_p$, being the $L_p$ obtained from the periodogram, based on the largest cycle for each time series [48] (i.e., about one trimester for all time series), being $N$ the length of the time series length. The number of eigentriples used for reconstruction $r$, for each of the considered window lengths and each of the time series, was obtained by analysing the w-correlations between components [23]. The number of eigentriples $r$ should be chosen in order to maximize the w-correlation among signal components, maximize the w-correlation among noise components, and minimize the w-correlation between signal and noise components, i.e., in order to maximize the separability between signal components and noise components.

**Table 3.** Window length $L_1 = N/20$, $L_2 = N/2$ and $L_p$, and number of eigentriples $r$ considered for model fit and model forecast for each of the currency exchange rates.

| Currency Exchange Rate | $L_1$ | $r_1$ | $L_2$ | $r_2$ | $L_p$ | $r_p$ | $L_{H-MSSA}$ | $r_{H-MSSA}$ | $L_{V-MSSA}$ | $r_{V-MSSA}$ |
|---|---|---|---|---|---|---|---|---|---|---|
| Brazilian real (USD/BRL) | 212 | 11 | 2120 | 7 | 60 | 20 | 60 | 30 | 60 | 21 |
| Chinese renminby (USD/CNY) | 212 | 11 | 2120 | 7 | 60 | 18 | 60 | 30 | 60 | 21 |
| Euro (USD/EUR) | 212 | 12 | 2120 | 14 | 60 | 13 | 60 | 30 | 60 | 21 |
| British pound (USD/GBP) | 212 | 10 | 2120 | 19 | 60 | 10 | 60 | 30 | 60 | 21 |
| Indian rupee (USD/INR) | 212 | 10 | 2120 | 7 | 60 | 17 | 60 | 30 | 60 | 21 |
| Japanese yen (USD/JPY) | 212 | 7 | 2120 | 10 | 60 | 16 | 60 | 30 | 60 | 21 |
| Russian rouble (USD/RUB) | 212 | 9 | 2120 | 7 | 60 | 15 | 60 | 30 | 60 | 21 |
| South African rand (USD/ZAR) | 212 | 8 | 2120 | 11 | 60 | 15 | 60 | 30 | 60 | 21 |

Figure 3 shows the w-correlation matrices for each of the eight currency exchange rates, considering the window length $L_p$ obtained based on the periodogram. Figures A1 and A2 of the appendix show the w-correlation matrices for each of the eight currency exchange rates, considering the window $L_1 = N/20$ and $L_2 = N/2$, respectively. Figure A3 of the appendix show the w-correlations for the horizontal and vertical MSSA. These w-correlation plots can be obtained with the function "wcor" of the R package "Rssa" [49]. These w-correlation plots intend to help with the decision about the separability between signal and noise components (3rd step of the SSA and MSSA algorithms). Being the darker colors of Figure 3 associated with higher w-correlations and lighter colors associated with lower w-correlations, we intend to choose the "best" cut-point that maximizes the separability, i.e., high w-correlations between signal components, high w-correlations between noise components, and low w-correlations between signal and noise components.

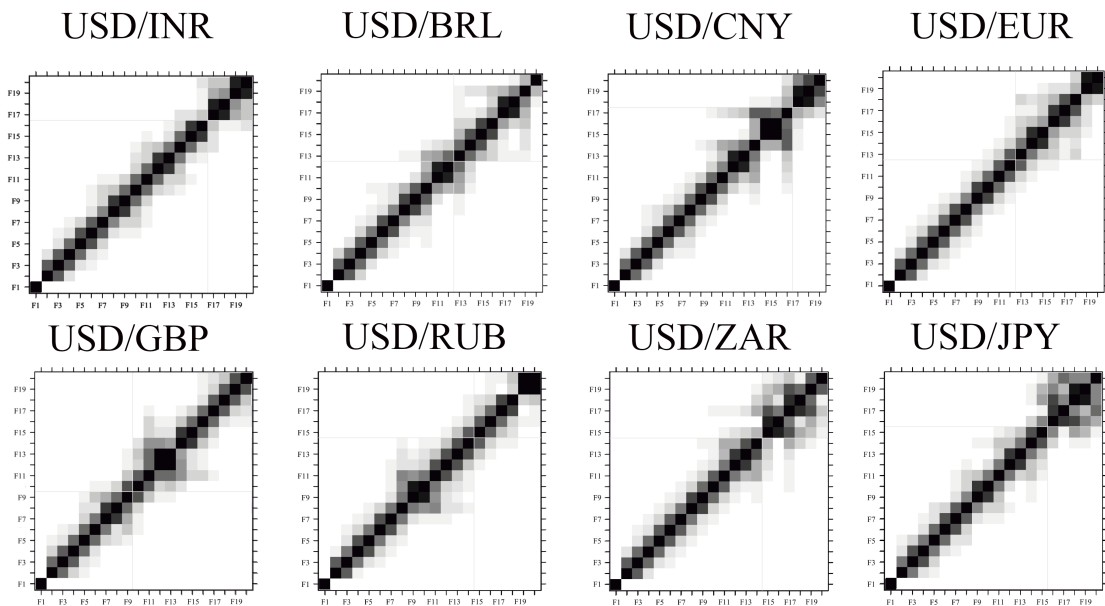

**Figure 3.** W-correlation matrices for each of the eight currency exchange rates, considering an window length $L_p$. The vertical and horizontal lines in each w-correlations plot indicate the selected cut-point that maximize separability between signal and noise components.

To access and compare the ability for model fit, the RMSE and the MAPE were calculated for each of the eight models/algorithms, one ARIMA, three SSA, two MSSA, one ANN, and one hybrid SSA-ANN, in each time series (Table 4 and Table 5, respectively). The results for the univariate and multivariate SSA are for the parameters defined in Table 3. The parameter for the SSA part of the hybrid method that combines SSA and ANN were chosen to be $L_p$ and $r_p$ (Table 3) because of the best fit when compared with the other parameter choices for SSA.

The results in Tables 4 and 5 show that the overall best SSA algorithm to fit the time series was the one with parameter $L_p$ and $r_p$ (Table 3), which also outperformed the ARIMA model and, in most cases, the ANN. The best multivariate SSA algorithm was the one that uses the horizontal form of the trajectory matrix (H-MSSA), that also outperformed all SSA algorithms and the ANN. However, the best overall model for model fit in the considered eight time series of exchange currency rates, was the hybrid model that combined the SSA and the ANN.

**Table 4.** Root mean square error for model fit for each of the eight currency exchange rates, considering each of the eight models/algorithms, ARIMA, SSA for the window length and number of eigentriples for reconstruction as defined in Table 3, multivariate SSA for the window length and number of eigentriples for reconstruction as defined in Table 3, ANN, and the hybrid method that combines SSA and ANN.

| Currency Exchange Rate | $ARIMA$ | $SSA_{L_1}$ | $SSA_{L_2}$ | $SSA_{L_p}$ | $H-MSSA$ | $V-MSSA$ | $ANN$ | $SSA-ANN$ |
|---|---|---|---|---|---|---|---|---|
| Brazilian real (USD/BRL) | 0.0317 | 0.0293 | 0.1235 | 0.0215 | 0.0124 | 0.0545 | 0.0291 | 0.0074 |
| Chinese renminby (USD/CNY) | 0.0129 | 0.0144 | 0.0773 | 0.0072 | 0.0050 | 0.0259 | 0.0119 | 0.0020 |
| Euro (USD/EUR) | 0.0057 | 0.0062 | 0.0151 | 0.0039 | 0.0025 | 0.0332 | 0.0053 | 0.0016 |
| British pound (USD/GBP) | 0.0038 | 0.0055 | 0.0169 | 0.0028 | 0.0015 | 0.0301 | 0.0039 | 0.0013 |
| Indian rupee (USD/INR) | 0.2767 | 0.3013 | 1.2830 | 0.1738 | 0.1163 | 0.2988 | 0.2727 | 0.0498 |
| Japanese yen (USD/JPY) | 0.7668 | 1.1127 | 2.5112 | 0.5831 | 0.3389 | 0.6140 | 0.6146 | 0.1812 |
| Russian rouble (USD/RUB) | 0.4739 | 0.6480 | 2.2367 | 0.4268 | 0.1775 | 0.4763 | 0.3839 | 0.0849 |
| South African rand (USD/ZAR) | 0.1109 | 0.1466 | 0.3833 | 0.0769 | 0.0434 | 0.2023 | 0.1114 | 0.0255 |

**Table 5.** Mean absolute percentage error for model fit for each of the eight currency exchange rates, considering each of the eight models/algorithms, ARIMA, SSA for the window length and number of eigentriples for reconstruction as defined in Table 3, multivariate SSA for the window length and number of eigentriples for reconstruction as defined in Table 3, ANN, and the hybrid method that combines SSA and ANN.

| Currency Exchange Rate | $ARIMA$ | $SSA_{L_1}$ | $SSA_{L_2}$ | $SSA_{L_p}$ | $H-MSSA$ | $V-MSSA$ | $ANN$ | $SSA-ANN$ |
|---|---|---|---|---|---|---|---|---|
| Brazilian real (USD/BRL) | 0.82% | 0.78% | 3.71% | 0.57% | 0.32% | 1.50% | 0.75% | 0.21% |
| Chinese renminby (USD/CNY) | 0.11% | 0.14% | 0.80% | 0.06% | 0.05% | 0.27% | 0.10% | 0.02% |
| Euro (USD/EUR) | 0.47% | 0.58% | 1.45% | 0.33% | 0.20% | 3.34% | 0.45% | 0.13% |
| British pound (USD/GBP) | 0.43% | 0.64% | 2.05% | 0.32% | 0.17% | 3.67% | 0.44% | 0.15% |
| Indian rupee (USD/INR) | 0.33% | 0.38% | 1.81% | 0.22% | 0.14% | 0.38% | 0.33% | 0.06% |
| Japanese yen (USD/JPY) | 0.47% | 0.79% | 1.89% | 0.38% | 0.20% | 0.40% | 0.43% | 0.10% |
| Russian rouble (USD/RUB) | 0.52% | 0.77% | 3.49% | 0.48% | 0.21% | 0.57% | 0.50% | 0.12% |
| South African rand (USD/ZAR) | 0.80% | 1.09% | 2.98% | 0.56% | 0.31% | 1.55% | 0.80% | 0.19% |

Table 6 shows the computational time for each of the eight currency exchange rates, considering each of the eight models/algorithms, ARIMA, SSA for the window length and number of eigentriples for reconstruction as defined in Table 3, multivariate SSA for the window length and number of eigentriples for reconstruction as defined in Table 3, ANN, and the hybrid method that combines SSA and ANN. It can be seen that, although the hybrid algorithm that combines the SSA and ANN takes longer that the competing methods, the computational times are under three minutes.

**Table 6.** Computational time, in minutes, for model fit, for each of the eight currency exchange rates, considering each of the eight models/algorithms, ARIMA, SSA for the window length and number of eigentriples for reconstruction as defined in Table 3, multivariate SSA for the window length and number of eigentriples for reconstruction as defined in Table 3, ANN, and the hybrid method that combines SSA and ANN.

| Currency Exchange Rate | $ARIMA$ | $SSA_{L_1}$ | $SSA_{L_2}$ | $SSA_{L_p}$ | H-MSSA[1] | V-MSSA[1] | $ANN$ | $SSA-ANN$ |
|---|---|---|---|---|---|---|---|---|
| Brazilian real (USD/BRL) | 0.3847 | 0.0360 | 0.8881 | 0.0132 | 0.0133 | 0.0923 | 0.1048 | 2.4027 |
| Chinese renminby (USD/CNY) | 0.2677 | 0.0290 | 0.8860 | 0.0172 | 0.0133 | 0.0923 | 0.1038 | 1.8119 |
| Euro (USD/EUR) | 0.2218 | 0.0281 | 0.9392 | 0.0145 | 0.0133 | 0.0923 | 0.2378 | 2.7058 |
| British pound (USD/GBP) | 0.0712 | 0.0259 | 0.9157 | 0.0139 | 0.0133 | 0.0923 | 0.0806 | 2.6205 |
| Indian rupee (USD/INR) | 0.1378 | 0.0412 | 0.8880 | 0.0186 | 0.0133 | 0.0923 | 0.1804 | 2.2644 |
| Japanese yen (USD/JPY) | 0.1970 | 0.0223 | 10.111 | 0.0112 | 0.0133 | 0.0923 | 1.6194 | 2.6516 |
| Russian rouble (USD/RUB) | 0.1064 | 0.0305 | 0.8561 | 0.0105 | 0.0133 | 0.0923 | 0.8474 | 2.6506 |
| South African rand (USD/ZAR) | 0.0859 | 0.0300 | 1.0156 | 0.0146 | 0.0133 | 0.0923 | 0.0746 | 1.6494 |

[1] The reported times are to obtain the results for the eight time series together.

Figure 4 shows the original time series, the smoothed time series after applying the SSA considering a window length $L_p$ and $r_p$ eigentriples (Table 3) and the model fit by the hybrid algorithm that combines the SSA and the ANN, for each of the eight currency exchange rates. It can be seen that the model fits almost overlap with the original time series, which was expected because of the overall low values for the RMSE (Table 4) and MAPE (Table 5). Similar behaviour was obtained by all considered methods.

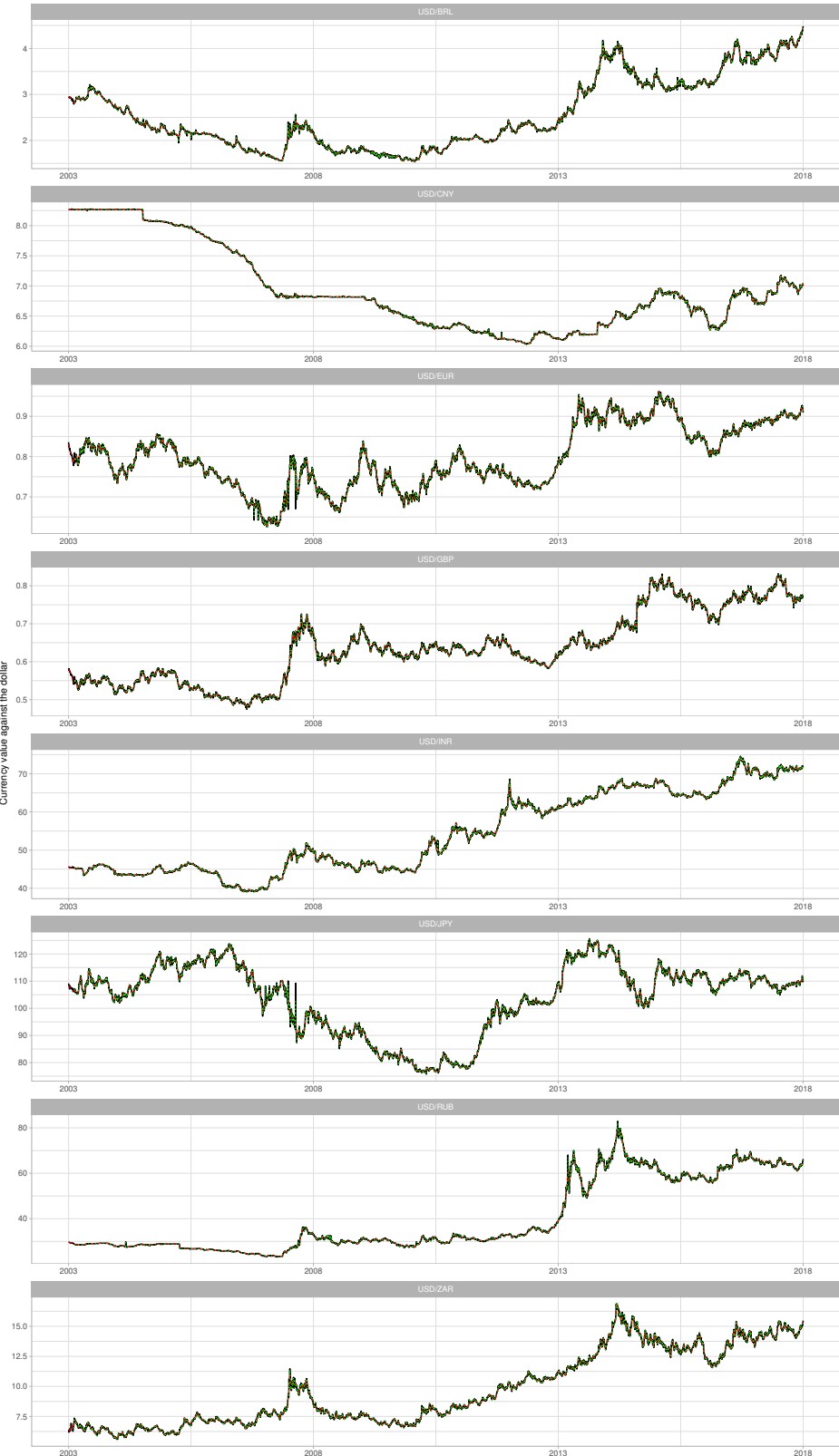

**Figure 4.** Original time series (black line), smoothed time series after applying the SSA considering a window length $L_p$ and $r_p$ eigentriples (Table 3) (red line) and model fit by the hybrid algorithm that combines the SSA and the ANN (green line), for each of the eight currency exchange rates. From top to bottom: USD/BRL, USD/CNY, USD/EUR, USD/GBP, USD/INR, USD/JPY, USD/RUB, and USD/ZAR. The vertical axes show the exchange rate and the horizontal axes shows the time.

### 3.2. Model Forecasting

In this section, we compare the forecasting ability of the eight models/algorithms under study: (i) ARIMA; (ii) SSA (with three alternative parameter choice); (iii) multivariate SSA (two algorithms); (iv) artificial neural networks (ANN); and (v) hybrid algorithm that combines SSA and ANN. Tables 7 and 8 give the RMSE and MAPE forecasting values, respectively, for each method/algorithm applied to each time series. These values are obtained by considering a test set of $g = 12$ observations from each time series, obtained for one, five and ten steps ahead out-of-sample forecast, i.e., one day ahead, one week ahead and two weeks ahead. The overall best performance, based on both RMSE and MAPE was obtained by the hybrid algorithm that combines the SSA and the ANN, for any number of steps ahead out-of-sample forecasts. For one-step-ahead out of sample forecasting, the second best overall performance was obtained by the SSA based on $L_p$ and $r_p$ (Table 3) and the multivariate SSA algorithms (Tables 7 and 8). For five and ten steps-ahead out of sample forecasting, the models ARIMA, SSA with $L_1 = N/20$ and $r_1$, SSA with $L_p$ and $r_p$, both versions of the multivariate SSA algorithm and the ANN, perform similarly in terms of RMSE (Table 7). When considering the MAPE (Table 8 for five and ten steps-ahead out of sample forecasting, the second best performance alternates between the multivariate versions of the SSA algorithm, the ANN and the SSA algorithms based on $L_p$ and $r_p$, and based on $L_1$ and $r_1$ (Table 3).

Although Tables Tables 7 and 8 only give the point estimates for the RMSE and MAPE, respectively, a measure of variability such as the standard errors could also be obtained based on resampling. To reduce the variability in these measure, the test size could also be increased which, in this case, provides similar results.

**Table 7.** Root mean square error for model forecasting for each of the eight currency exchange rates, considering each of the eight models/algorithms, ARIMA, SSA for the window length and number of eigentriples for reconstruction as defined in Table 3, multivariate SSA for the window length and number of eigentriples for reconstruction as defined in Table 3, ANN, and the hybrid method that combines SSA and ANN.

| Currency Exchange Rate | ARIMA | $SSA_{L_1}$ | $SSA_{L_2}$ | $SSA_{L_p}$ | H-MSSA | V-MSSA | ANN | $SSA-ANN$ |
|---|---|---|---|---|---|---|---|---|
| | | | | one-step-ahead | | | | |
| Brazilian real (USD/BRL) | 0.1323 | 0.0370 | 0.2580 | 0.0372 | 0.0410 | 0.0348 | 0.0494 | 0.0247 |
| Chinese renminby (USD/CNY) | 0.0239 | 0.0183 | 0.1644 | 0.0148 | 0.0248 | 0.0135 | 0.0407 | 0.0091 |
| Euro (USD/EUR) | 0.0110 | 0.0095 | 0.0076 | 0.0038 | 0.0029 | 0.0037 | 0.0056 | 0.0017 |
| British pound (USD/GBP) | 0.0056 | 0.0030 | 0.0461 | 0.0037 | 0.0042 | 0.0035 | 0.0048 | 0.0026 |
| Indian rupee (USD/INR) | 0.3448 | 0.2141 | 2.5460 | 0.2521 | 0.2231 | 0.2069 | 0.2802 | 0.1498 |
| Japanese yen (USD/JPY) | 0.9702 | 0.8953 | 11.461 | 0.7099 | 0.6853 | 0.6515 | 0.7578 | 0.4927 |
| Russian rouble (USD/RUB) | 2.1820 | 0.9059 | 1.5589 | 0.4637 | 0.6613 | 0.5168 | 1.4898 | 0.2807 |
| South African rand (USD/ZAR) | 0.4340 | 0.2963 | 0.2387 | 0.1040 | 0.1273 | 0.1023 | 0.2165 | 0.0723 |
| | | | | five-steps-ahead | | | | |
| Brazilian real (USD/BRL) | 0.2280 | 0.0544 | 0.2727 | 0.0648 | 0.0788 | 0.0645 | 0.0804 | 0.0209 |
| Chinese renminby (USD/CNY | 0.0273 | 0.0179 | 0.1738 | 0.0282 | 0.0489 | 0.0303 | 0.0418 | 0.0078 |
| Euro (USD/EUR) | 0.0107 | 0.0124 | 0.0070 | 0.0087 | 0.0088 | 0.0094 | 0.0071 | 0.0025 |
| British pound (USD/GBP) | 0.0056 | 0.0124 | 0.0469 | 0.0044 | 0.0063 | 0.0056 | 0.0047 | 0.0025 |
| Indian rupee (USD/INR) | 0.3632 | 0.2212 | 2.5601 | 0.5689 | 0.4848 | 0.3977 | 0.2850 | 0.1209 |
| Japanese yen (USD/JPY) | 0.9709 | 0.9186 | 11.842 | 1.2772 | 1.5018 | 1.0590 | 0.7407 | 0.5181 |
| Russian rouble (USD/RUB) | 2.1820 | 1.1078 | 1.6070 | 0.9951 | 1.3190 | 1.1384 | 1.4981 | 0.2759 |
| South African rand (USD/ZAR) | 0.4340 | 0.3613 | 0.2444 | 0.2105 | 0.2776 | 0.2053 | 0.1728 | 0.0470 |
| | | | | ten-steps-ahead | | | | |
| Brazilian real (USD/BRL) | 0.3498 | 0.0891 | 0.2909 | 0.0941 | 0.0890 | 0.0974 | 0.1499 | 0.0232 |
| Chinese renminby (USD/CNY) | 0.0457 | 0.0481 | 0.1862 | 0.0516 | 0.0653 | 0.0415 | 0.0434 | 0.0080 |
| Euro (USD/EUR) | 0.0107 | 0.0157 | 0.0059 | 0.0134 | 0.0137 | 0.0138 | 0.0090 | 0.0023 |
| British pound (USD/GBP) | 0.0056 | 0.0066 | 0.0478 | 0.0070 | 0.0060 | 0.0044 | 0.0047 | 0.0024 |
| Indian rupee (USD/INR) | 0.3836 | 0.3020 | 2.5821 | 0.5738 | 0.5168 | 0.4507 | 0.3151 | 0.1011 |
| Japanese yen (USD/JPY) | 0.9709 | 0.9404 | 12.293 | 1.0223 | 1.2728 | 1.1189 | 0.8205 | 0.3122 |
| Russian rouble (USD/RUB) | 2.1820 | 1.5443 | 1.6784 | 0.8272 | 1.6145 | 1.2535 | 1.8017 | 0.2482 |
| South African rand (USD/ZAR) | 0.4340 | 0.3854 | 0.2467 | 0.2894 | 0.3442 | 0.2878 | 0.2019 | 0.0470 |

**Table 8.** Mean absolute percentage error for model forecasting for each of the eight currency exchange rates, considering each of the eight models/algorithms, ARIMA, SSA for the window length and number of eigentriples for reconstruction as defined in Table 3, multivariate SSA for the window length and number of eigentriples for reconstruction as defined in Table 3, ANN, and the hybrid method that combines SSA and ANN.

| Currency Exchange Rate | *ARIMA* | $SSA_{L_1}$ | $SSA_{L_2}$ | $SSA_{L_p}$ | *H-MSSA* | *V-MSSA* | *ANN* | *SSA − ANN* |
|---|---|---|---|---|---|---|---|---|
| | | | | one-step-ahead | | | | |
| Brazilian real (USD/BRL) | 2.82% | 0.64% | 5.84% | 0.71% | 0.76% | 0.62% | 1.02% | 0.51% |
| Chinese renminby (USD/CNY) | 0.31% | 0.20% | 2.32% | 0.15% | 0.29% | 0.14% | 0.48% | 0.09% |
| Euro (USD/EUR) | 1.12% | 0.91% | 0.77% | 0.32% | 0.26% | 0.31% | 0.53% | 0.13% |
| British pound (USD/GBP) | 0.58% | 0.33% | 5.95% | 0.39% | 0.47% | 0.40% | 0.49% | 0.31% |
| Indian rupee (USD/INR) | 0.37% | 0.25% | 3.54% | 0.31% | 0.28% | 0.24% | 0.35% | 0.17% |
| Japanese yen (USD/JPY) | 0.58% | 0.52% | 10.35% | 0.45% | 0.49% | 0.40% | 0.56% | 0.39% |
| Russian rouble (USD/RUB) | 3.07% | 1.02% | 2.18% | 0.52% | 0.83% | 0.63% | 1.99% | 0.38% |
| South African rand (USD/ZAR) | 2.66% | 1.62% | 1.19% | 0.54% | 0.71% | 0.55% | 1.27% | 0.37% |
| | | | | five-steps-ahead | | | | |
| Brazilian real (USD/BRL) | 5.10% | 1.08% | 6.17% | 1.17% | 1.44% | 1.22% | 1.52% | 0.39% |
| Chinese renminby (USD/CNY) | 0.32% | 0.20% | 2.46% | 0.37% | 0.58% | 0.35% | 0.50% | 0.08% |
| Euro (USD/EUR) | 1.10% | 1.21% | 0.70% | 0.83% | 0.82% | 0.90% | 0.71% | 0.23% |
| British pound (USD/GBP) | 0.58% | 0.41% | 6.05% | 0.43% | 0.66% | 0.63% | 0.49% | 0.31% |
| Indian rupee (USD/INR) | 0.39% | 0.24% | 3.56% | 0.66% | 0.60% | 0.42% | 0.36% | 0.12% |
| Japanese yen (USD/JPY) | 0.58% | 0.55% | 10.69% | 0.86% | 1.02% | 0.71% | 0.56% | 0.40% |
| Russian rouble (USD/RUB) | 3.07% | 1.42% | 2.25% | 1.33% | 1.75% | 1.36% | 2.01% | 0.36% |
| South African rand (USD/ZAR) | 2.66% | 1.94% | 1.22% | 1.22% | 1.51% | 1.20% | 0.92% | 0.26% |
| | | | | ten-steps-ahead | | | | |
| Brazilian real (USD/BRL) | 7.93% | 1.96% | 6.59% | 1.93% | 1.80% | 2.07% | 3.27% | 0.46% |
| Chinese renminby (USD/CNY) | 0.56% | 0.55% | 2.64% | 0.69% | 0.79% | 0.53% | 0.52% | 0.09% |
| Euro (USD/EUR) | 1.10% | 1.61% | 0.56% | 1.28% | 1.32% | 1.32% | 0.91% | 0.21% |
| British pound (USD/GBP) | 0.58% | 0.77% | 6.17% | 0.73% | 0.61% | 0.49% | 0.50% | 0.34% |
| Indian rupee (USD/INR) | 0.42% | 0.35% | 3.59% | 0.64% | 0.58% | 0.52% | 0.41% | 0.11% |
| Japanese yen (USD/JPY) | 0.58% | 0.57% | 11.10% | 0.86% | 0.94% | 0.92% | 0.57% | 0.21% |
| Russian rouble (USD/RUB) | 3.07% | 2.26% | 2.36% | 0.95% | 2.06% | 1.54% | 5.20% | 0.30% |
| South African rand (USD/ZAR) | 2.66% | 2.08% | 1.23% | 1.54% | 2.00% | 1.49% | 0.99% | 0.25% |

The computational times to obtain the RMSE and MAPE values in Tables 7 and 8 are presented in Table 9. The lowest computational time was obtained for the multivariate SSA algorithms (the times reported in Table 9 are to obtain the forecast values for the eight time series together) in every number of steps ahead. These are followed by the SSA algorithms with $L_1$ and $L_p$ (Table 3) because of the more rectangular trajectory matrices used in the singular value decomposition, the most time consuming step of the SSA algorithm. As expected from the analysis of the computational times for model fit in Table 6, the hybrid model was the highest computational costly with times between 15 and 31 min, which was compensated with the excellent results in terms of model forecasting (Tables 7 and 8).

**Table 9.** Computational time, in minutes, for model fit, for each of the eight currency exchange rates, considering each of the eight models/algorithms, ARIMA, SSA for the window length and number of eigentriples for reconstruction as defined in Table 3, multivariate SSA for the window length and number of eigentriples for reconstruction as defined in Table 3, ANN, and the hybrid method that combines SSA and ANN.

| Currency Exchange Rate | *ARIMA* | $SSA_{L_1}$ | $SSA_{L_2}$ | $SSA_{L_p}$ | *H-MSSA* [1] | *V-MSSA* [1] | *ANN* | *SSA − ANN* |
|---|---|---|---|---|---|---|---|---|
| | | | | one-step-ahead | | | | |
| Brazilian real (USD/BRL) | 4.3655 | 0.5333 | 17.699 | 0.1891 | 0.2323 | 0.2755 | 0.8877 | 28.337 |
| Chinese renminby (USD/CNY) | 3.7305 | 0.4544 | 18.158 | 0.2448 | 0.2323 | 0.2755 | 0.8795 | 15.076 |
| Euro (USD/EUR) | 2.9998 | 0.4562 | 17.755 | 0.1813 | 0.2323 | 0.2755 | 0.9253 | 29.917 |
| British pound (USD/GBP) | 0.9440 | 0.5494 | 17.549 | 0.1530 | 0.2323 | 0.2755 | 0.8994 | 29.427 |
| Indian rupee (USD/INR) | 1.8074 | 0.4961 | 17.675 | 0.2385 | 0.2323 | 0.2755 | 2.2634 | 27.408 |
| Japanese yen (USD/JPY) | 2.9255 | 0.4253 | 17.502 | 0.2316 | 0.2323 | 0.2755 | 20.062 | 30.499 |
| Russian rouble (USD/RUB) | 1.3511 | 0.6087 | 17.294 | 0.3392 | 0.2323 | 0.2755 | 4.2888 | 29.711 |
| South African rand (USD/ZAR) | 1.1148 | 0.6444 | 17.377 | 0.2995 | 0.2323 | 0.2755 | 0.8910 | 19.033 |

**Table 9.** *Cont.*

| Currency Exchange Rate | ARIMA | $SSA_{L_1}$ | $SSA_{L_2}$ | $SSA_{L_p}$ | H-MSSA [1] | V-MSSA [1] | ANN | SSA − ANN |
|---|---|---|---|---|---|---|---|---|
| | | | five-steps-ahead | | | | | |
| Brazilian real (USD/BRL) | 4.5827 | 0.5158 | 17.352 | 0.1827 | 0.2112 | 0.2532 | 0.8935 | 28.563 |
| Chinese renminby (USD/CNY) | 4.0223 | 0.4433 | 17.328 | 0.2371 | 0.2112 | 0.2532 | 0.8928 | 15.157 |
| Euro (USD/EUR) | 3.3133 | 0.4625 | 17.574 | 0.1825 | 0.2112 | 0.2532 | 0.9231 | 30.004 |
| British pound (USD/GBP) | 1.0444 | 0.4335 | 17.296 | 0.1478 | 0.2112 | 0.2532 | 0.8987 | 29.270 |
| Indian rupee (USD/INR) | 1.9946 | 0.5102 | 17.376 | 0.2310 | 0.2112 | 0.2532 | 2.1407 | 26.658 |
| Japanese yen (USD/JPY) | 2.8206 | 0.3862 | 17.437 | 0.2172 | 0.2112 | 0.2532 | 19.977 | 30.510 |
| Russian rouble (USD/RUB) | 1.7594 | 0.4156 | 17.064 | 0.2088 | 0.2112 | 0.2532 | 4.1824 | 29.583 |
| South African rand (USD/ZAR) | 1.5077 | 0.4542 | 17.382 | 0.2100 | 0.2112 | 0.2532 | 0.8782 | 19.019 |
| | | | ten-steps-ahead | | | | | |
| Brazilian real (USD/BRL) | 4.6075 | 0.5136 | 17.217 | 0.1833 | 0.2335 | 0.2865 | 0.8950 | 28.491 |
| Chinese renminby (USD/CNY) | 4.0933 | 0.4469 | 17.272 | 0.2396 | 0.2335 | 0.2865 | 0.8961 | 15.133 |
| Euro (USD/EUR) | 3.3071 | 0.4646 | 17.442 | 0.1862 | 0.2335 | 0.2865 | 0.9321 | 29.974 |
| British pound (USD/GBP) | 1.0325 | 0.4219 | 17.230 | 0.1477 | 0.2335 | 0.2865 | 0.9158 | 29.441 |
| Indian rupee (USD/INR) | 1.9575 | 0.4892 | 17.271 | 0.2319 | 0.2335 | 0.2865 | 2.1563 | 26.646 |
| Japanese yen (USD/JPY) | 2.8240 | 0.3899 | 17.338 | 0.2184 | 0.2335 | 0.2865 | 20.062 | 30.689 |
| Russian rouble (USD/RUB) | 1.5355 | 0.4161 | 16.986 | 0.2080 | 0.2335 | 0.2865 | 4.2013 | 29.597 |
| South African rand (USD/ZAR) | 1.2403 | 0.4578 | 17.209 | 0.2099 | 0.2335 | 0.2865 | 0.8795 | 19.178 |

[1] The reported times are to obtain the results for the eight time series together.

## 4. Discussion and Conclusions

In this paper, we compared standard and advanced, parametric and non-parametric, and univariate and multivariate models to access the ability for model fit and model forecasting. The models under consideration were: (i) the ARIMA model; (ii) the univariate SSA model, considering three different choices for the window length $L$ and the number of eigentriples used for reconstruction $r$; (iii) the multivariate SSA model, considering the horizontal and vertical forms of the trajectory matrix and the linear recurrent algorithm; (iv) the ANN; and (v) a hybrid model that uses the SSA to fit/forecast the deterministic part of the data and the ANN to fit/predict the stochastic part of the data.

Based on previous analysis and comparisons, the non-parametric SSA proved to outperform standard methods such as the Holt–Winters and ARIMA models [38,50]. Another advantage of SSA in comparison with other standard methods for time series analysis and forecasting is that, contrary to those, it does not require the time series to be stationary. However, when the time series data includes outliers, the SSA which uses an SVD based on the least squares $L_2$ norm, might not be appropriated and gives worse results than a robust SSA algorithm which uses an SVD based on the $L_1$ norm [38,51]. For the case of multivariate time series data, the MSSA tends to outperform its univariate counterpart because as the co-integration between time series is considered in MSSA and not in SSA. The performance of MSSA for forecasting improves when there is dependency among time series [39]. Further developments in the field of time series forecasting have been obtained by combining different methods in hybrid methodologies which have proven to outperform most competing methods [9,52,53].

Although part of the initial motivation of using data on currency exchange rates from developing and developed countries, no specific similarity in behaviour was found nor specific interpretable cluster was obtained (Figure A4).

For both model fit and model forecasting, the best performance in terms of RMSE and MAPE was obtained by the hybrid method that combines the SSA and the ANN, although more expensive computationally. This was followed by the multivariate SSA algorithms with a much lower computational time. These results allow for possible further promising research directions such as the combination of the robust SSA algorithm [38,51,54] with ANN to model time series with data contamination with outlying observations, the combination of the randomized SSA algorithm [55] with ANN to reduce the computational time for long time series, and the combination of multivariate SSA algorithms [39] with ANN for multivariate time series analysis.

**Author Contributions:** Conceptualization, P.C.R. and O.O.A.; Formal analysis, P.C.R., J.S.P. and R.M.; Methodology, P.C.R., J.S.P. and R.M; Software, P.C.R., J.S.P. and R.M.; Supervision, P.C.R.; Visualization, J.S.P. and R.M.; Writing–original draft, P.C.R., O.O.A, J.S.P. and R.M.; Writing–review and editing, P.C.R., O.O.A, J.S.P. and R.M. All authors have read and agreed to the published version of the manuscript.

**Funding:** P.C.Rodrigues acknowledges financial support from the Brazilian National Council for Scientific and Technological Development (CNPq), grant number 305852/2019-1. O.O.Awe acknowledges financial support from the Brazilian Federal Agency for Support and Evaluation of Graduate Education (CAPES), under the program CAPES PrInt UFBA, grant number 88887.374271/2019-00.

**Conflicts of Interest:** The authors declare no conflict of interest.

## Abbreviations

The following abbreviations are used in this manuscript:

| | |
|---|---|
| ANN | artificial neural network |
| ARMA | autoregressive moving average |
| ARIMA | autoregressive integrated moving average |
| BRICS | Brazil, Russia, India, China, South Africa |
| BRL | Brazilian real |
| CNY | Chinese renminby |
| EUR | Euro |
| GBP | British pound |
| H-MSSA | horizontal form of the MSSA algorithm |
| INR | Indian rupee |
| JPY | Japanese yen |
| MAPE | mean absolute percentage error |
| MSSA | multivariate singular spectrum analysis |
| RUB | Russian rouble |
| SSA | singular spectrum analysis |
| SVD | Singular value decomposition |
| RMSE | Root mean square error |
| USD | United States dollar |
| V-MSSA | vertical form of the MSSA algorithm |
| ZAR | South African rand |

## Appendix A

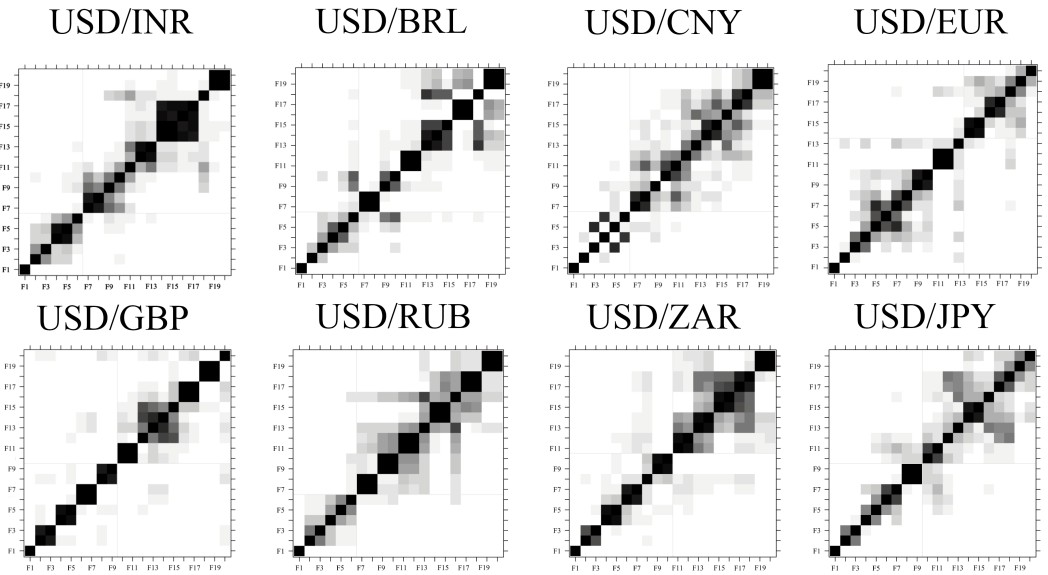

**Figure A1.** W-correlation matrices for each of the eight currency exchange rates, considering an window length $L_1 = N/20$. The vertical and horizontal lines in each w-correlations plot indicate the selected cut-point that maximize separability between signal and noise components.

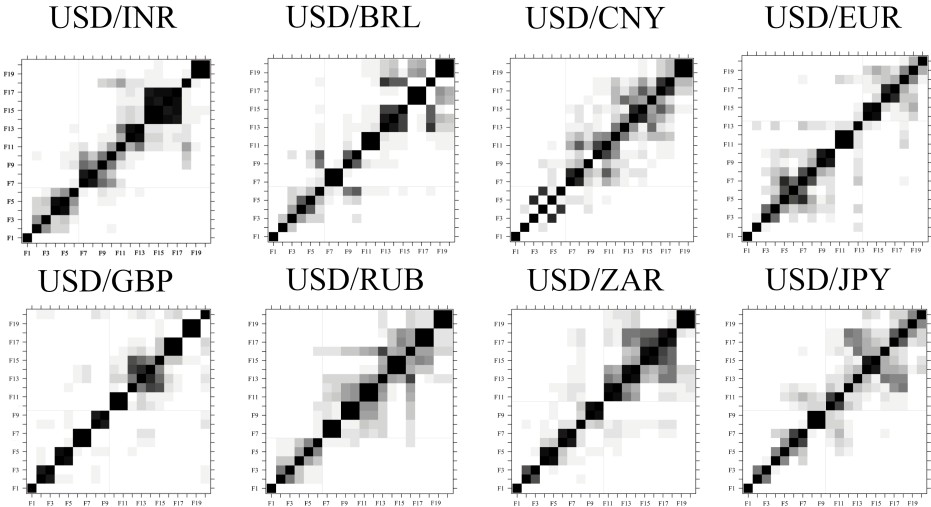

**Figure A2.** W-correlation matrices for each of the eight currency exchange rates, considering an window length $L_2 = N/2$. The vertical and horizontal lines in each w-correlations plot indicate the selected cut-point that maximize separability between signal and noise components.

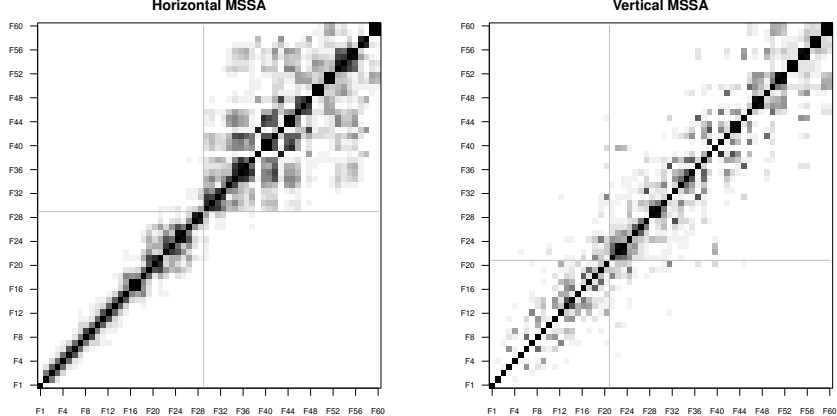

**Figure A3.** W-correlation matrices for the horizontal (H-MSSA; left hand side plot) and vertical (V-MSSA; right hand side plot) versions of the multivariate SSA that combines all eight currency exchange rate time series, considering window lengths of $L_{H-MSSA}$ and $L_{V-MSSA}$ (Table 3), respectively. The vertical and horizontal lines in each w-correlations plot indicate the selected cut-point that maximize separability between signal and noise components.

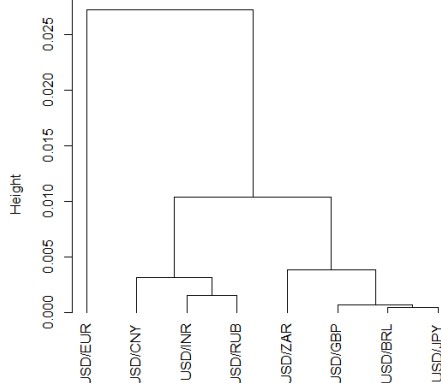

**Figure A4.** Dendrogram for the hierarchical cluster analysis for the eight currency, obtained using the "TSclust" package [56] of the R software.

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
