# Peer review of "Modelling the Behaviour of Currency Exchange Rates with Singular Spectrum Analysis and Artificial Neural Networks"

_stats, doi:10.3390/stats3020012_

Round 1

Reviewer 1 Report

This report is written after reading the paper in the evaluation, checking the references, and looking into them.

The manuscript uses different statistical and machine learning approaches to compare standard/advanced parametric and non-parametric, univariate/multivariate models to access the ability for model fit and model forecasting of the behavior of currency exchange rates.

The article is well-structured. The writing and organization of the paper are ok for the reviewer. The importance and relevance are in the scope of the Journal. The clarity of the paper is good. The references are enough.

Describing and Discussing Results

Nonstationarity:

It is essential that the authors take on this issue in more detail in the body of the text; they should provide evidence of stationarity and seasonality. The use of non-stationary data can lead to spurious regressions and that the persistence of a shock may be infinite.

It is essential that the authors address this issue for Series Prediction in more detail in the body of the text. They should provide evidence of stationarity. The use of non-stationary data can lead to spurious regressions and that the persistence of a shock may be infinite.

1) The reason for large variance over folds could be understood as an overfitting problem and small-size of the dataset. In Tables 4 and 5, it would be better to show the total standard error of these metrics (Confidence interval).

2) For evaluating and selecting the model, it should be done with cross validation (double or nested cross-validation) in order to generalize for a future set of data.

Quality-of-Fit Estimators:

The authors use one quality-of-fit summary estimators, RMSE for Model evaluation, but it would be helpful to have another measure like MAPE in the assessment process as stated in Hyndman, R. J. (2006). "Another look at measures of forecast accuracy", FORESIGHT Issue 4 June 2006, pg46. It would be also useful to assert the choices of n for the test set size. With autoregression on time series, it is often possible to get good fits with ARIMA for targets with in-range times but for the forecasts to be poor.

To add more quality to this work, I suggest the authors add a Discussion Section to argue pros and cons they find when compared with other techniques. A thorough review of its content has been performed and the referee hopes that, above all, it gives constructive comments.

Reviewer 2 Report

Main comments

  1. 4, Figure 1: This figure is rather incomplete. For example, why are some, but not all, of the weights labelled? In addition, in the accompanying discussion, the Z’s are not mentioned. (Should they be subscripted, 1, 2, 3?)
  2. 4, equation (3): no subscripts appear on the X and W variables. What is “g”, and how do we get from “g” to “f”? The discussion is incomplete.
  3. RMSE is used as the sole basis for measuring the quality of model fit and forecasting performance. Are the results sensitive to this choice? For example, what if Mean Absolute Error were used – are the same conclusions reached?
  4. Does it make sense to use a symmetric loss function? In other words, is it the case that under-prediction of the exchange rate (as currently expressed in USD) is just as important as over-prediction?
  5. The results in Table 4 are not particularly helpful unless we are told the details of the computer that was used – g., processor speed, number of processors, RAM, operating system, etc.
  6. 12, Figure 4: This figure is essentially illegible, and needs to be presented in a more readable way.

Minor comments

  1. Why is equation (2) even needed?
  2. 3, line 83: What studies? References are needed.
  3. 3, line 115: A reference is need for Rosenblatt and Widrow.
  4. 8, equation (12): Define “N”.
  5. 8, equation (13): Is the range of summation correct, given the way that “g” and “h” are defined?
  6. 9, Table 1: Even though the exchange rates have the same units (USD), they have different means, and so it would make more sense to report the coefficients of variation rather than the standard deviations
  7. 10, line 287: We are given absolutely no explanation about what the w-correlation matrices tell us or how they are to be interpreted.
  8. There are numerous spelling and grammatical errors that need to be addressed. For example:

p.1, line 22: “….in the literature”

p.4, line 145: “…this relatively new…..”

p.5, line 169: “hankelized” should be “Hankelized”.

p.7, line 213: “First”.

p.7, line 219: “….a unitary matrix……”.

p.7, lines 227 & 230: “de-noised” (or “smoothed”????).

p.8, lines 245 & 250: “…. associated to ....” should be “…. associated with …”.

p.8, line 250: “forecast” should be “forecasting”.

Reviewer 3 Report

The paper is based on a standard horse race between different models.
Neural networks have been considered in the past and SSA is not new.

Round 2

Reviewer 3 Report

I am satisfied with the revised version